# Transcription-Coupled Repair and R-Loop Crosstalk in Genome Stability

**DOI:** 10.3390/ijms26083744

**Published:** 2025-04-16

**Authors:** Jeseok Jeon, Tae-Hong Kang

**Affiliations:** Department of Biomedical Sciences, Dong-A University, Busan 49315, Republic of Korea; 2577615@donga.ac.kr

**Keywords:** transcription-coupled repair, R-loop, genome stability, neurodegenerative disorders, cancer treatment

## Abstract

Transcription-coupled repair (TCR) and R-loops are two interrelated processes critical to the maintenance of genome stability during transcription. TCR, a specialized sub-pathway of nucleotide excision repair, rapidly removes transcription-blocking lesions from the transcribed strand of active genes, thereby safeguarding transcription fidelity and cellular homeostasis. In contrast, R-loops, RNA–DNA hybrid structures formed co-transcriptionally, play not only regulatory roles in gene expression and replication but can also contribute to genome instability when persistently accumulated. Recent experimental evidence has revealed dynamic crosstalk between TCR and R-loop resolution pathways. This review highlights current molecular and cellular insights into TCR and R-loop biology, discusses the impact of their crosstalk, and explores emerging therapeutic strategies aimed at optimizing DNA repair and reducing disease risk in conditions such as cancer and neurodegenerative disorders.

## 1. Introduction

DNA transcription is a tightly regulated process essential for cellular function, development, and survival. However, the complexity of transcriptional regulation and its integration with genome maintenance pathways make it susceptible to errors, including DNA damage and conflicts with other cellular processes [1]. One critical mechanism ensuring transcriptional fidelity and genome stability is transcription-coupled repair (TCR), a sub-pathway of nucleotide excision repair (NER) that specifically targets lesions obstructing transcriptional elongation by RNA polymerase II (RNAPII) [2,3]. TCR was first identified through studies of Cockayne syndrome (CS), a disorder characterized by mutations in key TCR genes such as CS-A and CS-B [4]. The mechanism involves recognition of RNAPII stalling at damaged DNA sites, recruitment of NER factors, lesion excision, and subsequent DNA synthesis [5]. Beyond damage repair, TCR coordinates with chromatin remodelers and transcription factors, emphasizing its multifaceted role in transcriptional regulation and genomic homeostasis [6,7].

Concurrently, R-loops, structures formed during transcription when nascent RNA hybridizes with the DNA template strand, leaving the non-template strand exposed, have emerged as critical regulators of genome function [8]. Initially viewed as transcriptional by-products, R-loops are now recognized for their regulatory functions, including gene expression modulation, chromatin organization, and DNA replication initiation. However, persistent or aberrantly accumulated R-loops pose significant threats to genome stability, acting as obstacles to transcription and replication machinery and contributing to DNA double-strand breaks [9].

The interplay between TCR and R-loops has recently gained attention as a crucial mechanism for maintaining transcriptional integrity [10]. R-loops can act as intermediates for DNA repair, but their persistence can also exacerbate transcriptional stress and genomic instability [11]. In this context, TCR appears to function as both a responder and regulator, resolving R-loop-induced transcriptional blocks while mitigating associated DNA damage. Studies have shown that TCR components, such as CSB and XPG, are directly involved in resolving R-loops, highlighting functional overlaps between DNA repair and RNA processing pathways [12]. The functional implications of TCR-R-loop interactions extend to various cellular processes, including replication stress responses, transcriptional pausing, and chromatin remodeling. Hence, dysregulation of these processes has been linked to pathological conditions such as neurodegenerative diseases, cancer, and premature aging syndromes [13,14]. For example, mutations in CSA and CSB, key TCR proteins, result in Cockayne syndrome, characterized by neurodevelopmental defects and increased sensitivity to DNA damage [15]. Similarly, elevated R-loop accumulation has been observed in cancers, where it contributes to replication stress and genomic instability, providing potential therapeutic targets [16,17].

In this review, we will examine the molecular players involved in these processes, highlight emerging therapeutic strategies, and discuss future research directions to address unresolved questions. By integrating findings from molecular, cellular, and computational studies, we will seek to illuminate the intricate crosstalk between transcription and repair pathways and its relevance to human health.

## 2. Molecular Mechanisms of Transcription-Coupled Repair

TCR (often termed transcription-coupled nucleotide excision repair, TC-NER) is initiated when RNAPII stalls at a DNA lesion on the transcribed strand (Figure 1). In humans, the key sensor protein Cockayne syndrome B (CSB, also known as ERCC6) is rapidly recruited to the stalled RNAPII, acting as the first responder in TCR [18]. CSB is an ATP-dependent DNA translocase that binds upstream of RNAPII and can attempt to remodel or translocate DNA to push the polymerase past minor blocks [19]. If the blockage is insurmountable due to a bulky adduct like a cyclobutane pyrimidine dimer, CSB remains at the stalled RNAPII and triggers the assembly of the TCR machinery [20]. Cockayne syndrome A (CSA, ERCC8) is then recruited; CSA functions as part of an E3 ubiquitin ligase complex (the CSA-DDB1-CUL4 ubiquitin ligase) that ubiquitylates CSB and likely RNAPII itself [21]. This ubiquitination serves to remodel or mark the stalled transcription complex, triggering subsequent steps. Another factor, UVSSA (UV-stimulated scaffold protein A), is concurrently recruited and forms a complex with USP7. UVSSA’s primary role is to stabilize CSB and to serve as a platform to recruit the core NER complex TFIIH to the lesion [22]. Once TFIIH, a multi-subunit transcription/repair factor with DNA helicase subunits XPB and XPD, is in place at the stalled polymerase, it unwinds the DNA around the lesion using its 3′→5′ and 5′→3′ helicase activities, respectively [23]. This local unwinding allows for the verification of the damage and opens the DNA for repair. The subsequent steps of TCR mirror those of global-genome NER (GGR): the structure-specific endonucleases XPG (which cuts on the 3′ side of the lesion) and ERCC1-XPF (cuts on the 5′ side) incise the damaged strand, excising an ~22–30 nucleotide oligonucleotide containing the lesion [24]. During this process, the presence of TFIIH and associated factors helps to maintain the open bubble and coordinate the incision. The excised oligomer (carrying the damage) is displaced and bound by TFIIH until it is released [25]. Then, DNA polymerase δ/ε, along with auxiliary factors, fills in the single-stranded gap using the undamaged strand as a template, and DNA ligase seals the nick to complete repair [26]. After repair, a restored duplex DNA is present, and RNAPII can resume transcription elongation from the repaired template [27]. Notably, if the lesion is repaired quickly, RNAPII may simply restart; in other cases, extensive ubiquitination of the largest subunit of RNAPII can signal its proteasomal degradation to clear the road for a new polymerase to re-initiate transcription [28]. Thus, TCR ensures that DNA lesions do not indefinitely block gene expression.

Multiple protein factors tightly regulate TCR. CSB itself is a chromatin remodeler and can recruit histone acetyltransferases to loosen chromatin around the lesion, facilitating repair access [29]. CSA, as part of the ubiquitin ligase complex, also helps to remove or remodel stalled transcription complexes via targeted protein degradation or conformational change [18]. By binding USP7 (a deubiquitinase), UVSSA counteracts the premature degradation of CSB, thereby balancing ubiquitination levels during repair [30]. In addition to these core factors, recent research has identified ELOF1 (elongation factor 1) and a protein kinase STK19 as important contributors to TCR efficiency [31]. Geijer et al. showed that ELOF1 promotes the recruitment of UVSSA and TFIIH to efficiently repair transcription-blocking lesions and resume transcription [32]. Additionally, Weegen et al. showed that ELOF1 modulates RNAPII ubiquitination, a key signal for the assembly of downstream repair factors, thereby preserving genome stability [33]. STK19 directly interacts with RNAPII, CSA, and TFIIH and is recruited to damage sites after CSA, thereby stimulating TCR efficiency [34]. Heuvel et al. showed that loss of STK19 does not impact initial TCR complex assembly or RNAPII ubiquitination but delays lesion-stalled RNAPII clearance, thereby interfering with the downstream repair reaction [35]. Using cryogenic electron microscopy, Ramadhin et al. demonstrated that STK19 is an integral part of the RNAPII-TCR complex, bridging CSA, UVSSA, RNAPII, and downstream DNA [36].

Overall, the molecular mechanism of TCR is a carefully coordinated process: RNAPII arrest at damage is sensed and converted into a call for repair, leveraging NER enzymes targeted specifically to the site of transcription blockage. This protects cells from the deleterious consequences of stalled transcription and ensures that genetic information can be continuously and accurately expressed even in the face of DNA damage.

## 3. Biogenesis and Regulation of R-Loops

R-loop formation is inherently tied to the process of transcription. As RNAPII elongates, it generates transient opportunities for the nascent RNA to anneal with the DNA template behind the transcription bubble (Figure 2). Normally, the nascent RNA is quickly sequestered by RNA-binding proteins, spliced, packaged into RNPs, and exported, which prevents hybridization. However, when certain conditions are met, such as a pause in transcription, specific DNA sequences, or torsional stress in DNA, the RNA can invade the DNA duplex behind the polymerase [10]. Sequences with a high GC-skew strongly favor R-loop formation because an RNA:DNA hybrid with many rG-dC pairs is thermodynamically more stable than the equivalent DNA:DNA duplex [37]. Many R-loop hotspots, such as promoter regions of genes, have this GC skew and a run of poly(dT) on the coding strand, which facilitates the RNA’s re-hybridization into the upstream duplex [38]. Furthermore, negative supercoiling behind a moving RNAPII promotes R-loops, which makes strand unwinding easier and can be relieved by the formation of an RNA:DNA hybrid [39]. In bacteria, it was shown that loss of DNA topoisomerase I, which normally relaxes negative supercoils, leads to rampant R-loop formation, as the cell tries to use R-loops as “topological sinks” to absorb the supercoiling stress [40]. Thus, R-loops naturally occur at genomic regions with the right combination of sequence propensity, RNA abundance, and DNA topology. Historically considered rare accidents, it is now clear that R-loops can form genome-wide under normal conditions, with tens of thousands of R-loop sites documented in human and mouse genomes [41]. R-loops often occur at gene regulatory regions like promoters and terminators, where they can influence chromatin structure and transcription output, and at repetitive elements such as immunoglobulin switch regions or rDNA spacers, where they participate in specialized functions [42]. Notably, recent studies have also reported that the CRISPR/Cas gene editing system can induce R-loop formation during target DNA recognition and cleavage as part of its mechanism for creating site-specific double-strand breaks [43]. Since uncontrolled R-loops can interfere with essential processes, cells have evolved multiple layers of R-loop regulation to ensure these structures are resolved or prevented when necessary. One major class of R-loop regulators are ribonucleases H (RNase H1 and H2). These enzymes specifically recognize RNA:DNA hybrids and degrade the RNA strand of the hybrid [44]. By cleaving the RNA in an R-loop, RNase H enzymes dismantle the R-loop structure, allowing the DNA duplex to re-anneal and eliminating the displaced ssDNA loop. They are considered a primary defense against persistent R-loops; indeed, overexpression of RNase H1 can rescue many cellular defects associated with excess R-loops, and conversely, RNase H loss leads to R-loop accumulation and genome instability [45].

A second critical group of regulators are helicases, which can actively unwind RNA–DNA hybrids. Many helicases with roles in DNA/RNA metabolism have been implicated in R-loop resolution. For example, Senataxin (SETX) is a 5′→3′ helicase that resolves R-loops, especially during transcription termination; mutations in SETX cause AOA2 (ataxia oculomotor apraxia 2) and ALS4, disorders marked by R-loop accumulation in neurons [46,47]. Other helicases such as Aquarius (AQR), the RECQ family members WRN and BLM, FANCM, DDX family RNA helicases (e.g., DDX5, DDX1, DDX21, and DHX9), and PIF1, ATRX, and RTEL1 have all been shown to unwind R-loops [45]. Some of these function in specific contexts, for instance, ATRX and RTEL1 suppress R-loops at telomeres and centromeres to maintain stability [48], and FANCM can dismantle R-loops at stalled replication forks to prevent collapse [49]. Intriguingly, a few helicases can promote R-loop formation as well: for example, in bacteria, the Cas3 helicase/nuclease helps create R-loops during CRISPR interference [50], and the human UPF1 helicase involved in nonsense-mediated decay can stabilize R-loops in certain contexts [51]. This highlights that helicases can have dual roles, by either unwinding or annealing nucleic acids, to modulate R-loops as needed. Indeed, helicases such as DDX1, DDX17, and DHX9 are involved in both R-loop formation and resolution [45].

Beyond RNases and helicases, topoisomerases are important indirect regulators of R-loops. As noted, Topoisomerase I (TopI) relieves negative supercoils during transcription; without TopI, R-loops are more likely to form, demonstrated by the fact that RNase H overexpression can compensate for TopI loss in bacteria by removing R-loops [52]. Topoisomerase II has been shown to preserve some negative supercoils at gene ends in yeast, limiting R-loop formation in those regions [53]. In addition, a host of RNA-binding and RNA-processing factors act co-transcriptionally to suppress R-loops. The THO/TREX complex is involved in mRNA export, and various splicing factors coat the nascent transcript and prevent it from re-hybridizing with DNA [54]. For instance, the spliceosomal protein SRSF1 and mRNA export factors like ALYREF/UAP56 help package the nascent RNA into a messenger ribonucleoprotein (mRNP) particle, which physically hinders R-loop formation [55,56]. Loss of such factors often leads to R-loop accumulation as an indirect consequence of incomplete RNA packaging. Furthermore, DNA repair proteins also monitor R-loops: BRCA1 and BRCA2, well-known for their roles in homologous recombination, participate in R-loop suppression at transcription–replication conflict sites and at telomeres, preventing R-loop-driven breaks [57]. Fanconi anemia proteins FANCD2 and FANCI also bind R-loops at stalled forks and recruit enzymes like FANCM helicase or RNase H2 to resolve them, thus protecting against R-loop-induced fork collapse [58].

In summary, the cell employs a network of enzymes, e.g., nucleases (RNase H1/H2), helicases, and topoisomerases, along with RNA processing factors, to carefully regulate R-loops. This ensures that R-loops can form when beneficial (for instance, to facilitate certain biological reactions or regulatory events) but are promptly removed if they become excessive or harmful. The balance maintained by these factors is critical, as a shift toward too many or persistent R-loops can trigger DNA damage signaling and genome instability.

## 4. Interplay Between TCR and R-Loops

Given that TCR and R-loops both intersect with the transcription machinery, it is perhaps not surprising that they influence each other’s outcomes. Research in recent years has illuminated a bidirectional relationship: on the one hand, TCR factors can affect the formation and resolution of R-loops; on the other hand, R-loops can modulate the activity of TCR or even serve as intermediates in certain repair processes (Figure 3) [39]. This interplay is complex, with both cooperative and deleterious interactions reported.

### 4.1. TCR Factors as a Source of Genome Instability via R-Loop Processing

Sollier et al. discovered that TCR factors such as CSB can sometimes inadvertently promote genome instability by processing R-loops as if they were DNA damage [59]. The accumulated R-loops due to inhibition of TopI or loss of certain RNA processing helicases like AQR or SETX are recognized and cleaved by the TCR machinery. Additionally, they also found that XPF–ERCC1 and XPG, the NER endonucleases, were found to incise R-loop structures, resulting in DNA double-strand breaks, and this processing required CSB, but did not require XPC, the key GGR factor. In other words, an R-loop that stalls RNAPII can recruit CSB and the TCR apparatus, leading to endonucleolytic cleavage of the DNA around the R-loop. This is analogous to how TCR would cut out a DNA lesion, except in this case, the “lesion” is an R-loop or associated abnormal structure. The outcome can be detrimental: concurrent cutting on both sides of an R-loop can generate a double-strand break. These findings revealed an unexpected pro-instability role for TCR factors in processing R-loops, essentially turning a transcription-blocking R-loop into a potentially lethal DSB (Figure 4A). However, this interplay suggests that the cell faces a dilemma: TCR factors acting on what they perceive as a “stalled transcription complex” might actually worsen the situation if the stall is due to an R-loop rather than a true DNA lesion. It provides one explanation for how the loss of RNA processing factors leading to R-loops can cause genome instability through the action of DNA repair nucleases.

### 4.2. Protective Roles of R-Loops as Cooperative Intermediates in TCR

Conversely, TCR factors can also play protective or cooperative roles with R-loops in certain contexts. An example of a positive interplay is seen in telomeres under oxidative stress [60]. Oxidative DNA damage at telomeres can induce the formation of R-loops through the telomeric repeat-containing RNA, TERRA, hybridizing to telomeric DNA. These R-loops, rather than being merely harmful, were found to recruit CSB to the damaged telomeres, along with the homologous recombination protein RAD52 [60]. CSB physically interacts with RAD52 and DNA:RNA hybrids, helping localize RAD52 to the R-loop. RAD52, in turn, recruits POLD3, and together they execute a form of break-induced replication (BIR) to repair the oxidative DNA double-strand breaks at the telomere. In this pathway, the presence of an R-loop actually promotes repair: the R-loop acts as a signal to gather a repair complex (CSB–RAD52–POLD3) that can heal the break using a homologous recombination-like process. Both CSB and RAD52 were shown to be required for efficient repair of these telomeric breaks, highlighting a cooperative role where a TCR factor CSB and an R-loop engage the homologous repair machinery to protect chromosome ends (Figure 4B). This is an elegant illustration that R-loops are not universally detrimental; in specific scenarios, they can serve as critical intermediates that TCR factors recognize to facilitate an alternate repair route.

### 4.3. TCR-Mediated Suppression of R-Loop Accumulation

Additionally, there is evidence that TCR activity can suppress R-loop accumulation during transcription. Under normal conditions, as RNAPII encounters obstructions, CSB can help it bypass mild impediments or signals for quick repair. This reduces prolonged stalling of RNAPII, which is a situation that favors R-loop formation. In cells deficient in CSB, genome-wide R-loop mapping by R-ChIP revealed a significant increase in R-loop formation, particularly in long, transcriptionally active genes [38]. The R-loops in CSB-deficient cells tended to accumulate within introns and in regions with sequences prone to R-loop formation, suggesting that normally, CSB (and TCR by extension) helps traverse difficult-to-transcribe regions without forming R-loops. In the absence of functional TCR, RNAPII may stall more frequently or for longer durations, giving the nascent transcript time to hybridize with DNA. Thus, TCR factors like CSB and CSA can indirectly prevent R-loops by maintaining transcriptional throughput and promptly removing lesions or obstacles that would cause polymerase pausing. This concept is supported by the finding that many R-loops induced by CSB knockdown occur at sites of repetitive runs (T-run sequences after GC-rich tracts) that likely cause polymerase pausing; normally, TCR would help deal with these natural pause sites to avoid hybrid formation [38].

In summary, the interplay between TCR and R-loops is multifaceted. TCR factors can mistakenly process R-loops as damage, leading to nuclease-induced breaks and genome instability if R-loops are excessive. On the other hand, TCR components like CSB also collaborate with R-loops in certain DNA repair scenarios, such as at damaged telomeres, to promote the healing of breaks. Moreover, effective TCR helps minimize opportunities for R-loop formation by reducing prolonged transcriptional stalls. Both processes work in concert to prevent “transcription-associated genome instability”, a phenomenon where transcription itself becomes a source of DNA damage. Additionally, recent studies suggest that RNA helicases not only resolve R-loops but also recruit regulatory factors such as METTL3 to R-loop regions. DDX21-mediated localization of METTL3 enables m6A modification on nascent RNA, thereby contributing to R-loop resolution, transcription termination, and genome stability. This indicates that RNA helicases may function as active regulators of R-loop homeostasis (Figure 3) [61]. The cell must carefully regulate this interplay: too little TCR and R-loops may build up, but uncontrolled TCR nuclease activity on R-loops can also be harmful. Understanding this crosstalk is key, as it influences how cells respond to complex situations where transcription, DNA damage, and RNA:DNA hybrids all collide.

## 5. Pathological Consequences of Dysregulated TCR and R-Loops

Given their crucial roles in maintaining genome integrity during transcription, it is not surprising that defects in TCR or R-loop homeostasis are linked to human disease. Inherited mutations in TCR pathway genes cause severe developmental and neurological disorders, while chronic R-loop dysregulation is associated with neurodegeneration, premature aging, immunodeficiency, and cancer [62].

### 5.1. Cockayne Syndrome: A Prototypical Disorder Caused by Loss of TCR

Cockayne syndrome (CS) is a prototypical disorder caused by loss of TCR. Mutations in the CSA (ERCC8) or CSB (ERCC6) genes lead to CS, a rare recessive disease characterized by progressive multi-system neurodegeneration, growth failure, photosensitivity, and features of premature aging [63]. Unlike Xeroderma Pigmentosum (XP), which is another NER defect syndrome leading to skin cancer, CS patients typically do not develop cancers; instead, they exhibit developmental abnormalities, cachectic dwarfism, sensorineural hearing loss, vision problems, and severe neurologic impairment [64]. Cells from CS patients are hypersensitive to UV irradiation and other DNA-damaging agents that block transcription because they cannot effectively resume transcription after DNA damage due to the failure of TCR [65]. The CSB protein sits at the interface of transcription and repair; without it, transcription-blocking lesions persist, causing prolonged RNAPII stalling, which triggers apoptosis or cellular senescence rather than being repaired [66]. The consequence is widespread transcriptional stress and cell dysfunction. At the molecular level, CS cells accumulate DNA damage in active genes and exhibit transcriptional arrest in response to even mild DNA damage, explaining the extreme sensitivity and developmental deficits. Intriguingly, although CS patients do not tend to develop cancer, mouse models with CSB mutations do show an elevated cancer incidence [38], suggesting that in some contexts, TCR loss can be oncogenic, but in humans, the syndrome’s lethal developmental effects manifest before cancer can arise. Recent evidence indicates that CSB deficiency also leads to R-loop accumulation in cells, especially in long genes and genes with repetitive sequences [67]. The inability to restart transcription might create a permissive environment for R-loops to form, which could contribute to the neurological aspects of the disease via genomic instability or aberrant gene expression in neurons. In sum, CS underscores the importance of TCR in preventing transcription-associated damage; its absence causes a dramatic aging-like phenotype and neurodegeneration, likely through a combination of unrepaired DNA lesions and secondary R-loop stress.

### 5.2. R-Loop Dysregulation in Neurodegenerative Disorders

Neurodegenerative disorders have emerged as another category of diseases linked to R-loop dysregulation [13]. Neurons are highly transcriptionally active and long-lived, making them particularly vulnerable to cumulative transcription stress [68]. Ataxia with Oculomotor Apraxia Type 2 (AOA2) and a form of adolescent ALS (amyotrophic lateral sclerosis type 4) are caused by mutations in the Senataxin (SETX) gene, which encodes a helicase that resolves R-loops during transcription termination [69]. AOA2 is a recessive loss-of-function in SETX, leading to progressive cerebellar degeneration and ataxia, whereas ALS4 is caused by dominant mutations that likely confer a toxic gain-of-function or dominant-negative effect on SETX [47]. In both cases, R-loops accumulate abnormally in neuronal cells, presumably leading to DNA breaks or disrupted gene regulation that causes neurodegeneration. Another example is Fragile X syndrome (and Fragile X-associated tremor/ataxia syndrome), where CGG repeat expansions in the *FMR1* gene form stable R-loops that contribute to repeat instability and gene silencing, as well as triggering local DNA damage and aberrant DNA methylation [70]. Importantly, recent evidence suggests that R-loop-induced DNA double-strand breaks (DSBs) in FXS are not confined to the CGG repeat locus but can occur across the genome [71]. Spinocerebellar ataxia 31 is also linked to R-loop-forming repeat sequences in an intron [72]. Furthermore, defects in RNase H2, as seen in the autoimmune/neurodevelopmental disorder Aicardi–Goutières syndrome, may lead to the accumulation of RNA:DNA hybrids and an interferon-mediated inflammatory response [73]. In these neurodegenerative conditions, the inability to properly resolve R-loops or cope with transcription stress leads to DNA strand breaks, activation of DNA damage responses, and likely the loss of post-mitotic neurons over time [74]. Importantly, many of these disorders implicate proteins that interface with transcription, e.g., SETX, DNA/RNA helicases, RNase H2, or even TCR factors like CSB. The nervous system seems especially susceptible to defects in these factors, highlighting that persistent R-loops and transcription-blocking lesions are not well tolerated in neurons. The result can be genomic instability and chronic DNA damage signaling in neuronal cells, contributing to neurodegeneration and functional decline.

### 5.3. Cancer Predisposition Linked to TCR or R-Loop Deficiencies

Deficiencies in TCR or R-loop regulation are also linked to cancer predisposition and progression. Cells that cannot efficiently repair transcription-blocking lesions may accumulate mutations in active genes or suffer DNA breaks [75]. For instance, a hallmark of many cancers is a mutation bias: because TC-NER normally repairs the transcribed strand, that strand accumulates fewer mutations; if TCR is compromised, this strand-specific protection is lost [21]. Mutational signatures in cancer genomes often reflect active TCR, e.g., in melanomas, UV-induced mutations (C→T at dipyrimidines) are found more on the non-transcribed strand, indicating TCR repaired many on the transcribed strand [76]. If a tumor had a defect in TCR, one might see more symmetric mutation patterns or mutations in normally protected genes. Indeed, rare individuals with combined features of XP, CS, and XP/CS complex syndromes have partial TCR defects and suffer both cancer and neurodegeneration [77]. Beyond mutations, R-loops themselves are a source of genomic fragility relevant to cancer [78]. Unresolved R-loops can lead to replication-transcription collisions, DNA breaks, and chromosome rearrangements, which are well-known drivers of tumorigenesis [79]. Several tumor suppressors and genome stability proteins are now understood to suppress R-loops: for example, BRCA1 and BRCA2 (breast/ovarian cancer suppressors) help resolve R-loops at stalled forks and DSB sites, and loss of BRCA1 leads to R-loop accumulation and replication stress [80]. Fanconi anemia proteins also act to restrain R-loops, especially at difficult-to-replicate regions, explaining why their absence causes chromosome breakage and fusion events [81]. Furthermore, cancers frequently exhibit mis-splicing or imbalances in RNA-processing factors, which can lead to heightened R-loop levels. For example, mutations in the spliceosome components like U2AF1, SRSF2 or SF3B1 in myelodysplastic syndromes and leukemia cause widespread splicing perturbations and R-loop accumulation [82]. These cancer cells become highly reliant on stress response pathways such as ATR-mediated checkpoint signaling to survive the R-loop onslaught. Therefore, inhibition of the ATR pathway using ATR inhibitors or RNA splicing perturbation by expression of the spliceosome mutant results in cancer cell death due to catastrophic DNA damage, indicating a potential vulnerability [82].

In summary, dysregulation of TCR and R-loops is a thread connecting diverse pathologies, from developmental neurodegenerative syndromes like CS or AOA2, where the primary problem is failure to handle transcription-coupled DNA damage or R-loops to cancer, where chronic transcriptional stress and R-loop-induced damage fuel genomic instability. Each of these conditions underscores that maintaining the balance between transcription, DNA repair, and R-loop resolution is essential for normal cellular and organismal health.

## 6. Therapeutic Implications

As our understanding of TCR and R-loops deepens, new opportunities are emerging to target these processes in disease contexts. Therapeutic strategies fall into two broad categories: (1) restoring or enhancing TCR and R-loop resolution in conditions where their dysfunction causes diseases such as genetic syndromes or neurodegeneration, and (2) exploiting vulnerabilities in TCR or R-loop pathways to treat diseases like cancer.

### 6.1. Gene Therapy and Enzyme Replacement

For genetic disorders caused by TCR or R-loop resolution deficiencies, a straightforward concept is gene therapy to introduce a functional copy of the defective gene. In CS, where patients lack CSA or CSB, delivering the correct gene to patient cells could restore TCR capability [83]. Similarly, in AOA2 or certain ataxias, providing a functional SETX helicase might reduce R-loop accumulation [84]. While technical hurdles, including delivery to neurons and ensuring adequate expression, are significant, the growing success of AAV-mediated gene therapy for neurological disorders is encouraging [85]. An alternative to full gene delivery is providing the missing enzyme activity; for instance, researchers have explored expressing RNase H1 in cells burdened with R-loops to directly digest the problematic RNA:DNA hybrids [86]. In model systems of neurodegeneration with excess R-loops, RNase H overexpression has been shown to mitigate DNA damage and improve cell survival [87]. In the future, targeted delivery of RNase H or specific helicases to affected tissues could be a strategy to alleviate diseases driven by R-loops. For example, one could imagine a tailored RNase H delivery to neurons in ALS4 or a small-molecule activator of a backup helicase to compensate for SETX loss [88]. These approaches are still in the early stages, but they offer a precision-medicine angle by supplementing the missing repair activity.

### 6.2. Small-Molecule Modulators

Another avenue is developing drugs that modulate TCR or R-loop levels. For diseases of R-loop excess, one might screen for small molecules that stabilize the genome by reducing R-loops [89]. This could include compounds that enhance the recruitment or activity of RNase H or helicases at R-loops or drugs that promote transcription elongation, thereby minimizing pauses and R-loop formation [90]. On the flip side, in cancer therapy, it may be desirable to increase R-loop stress in cancer cells to push them over the edge [91]. Certain chemotherapeutic drugs already do this: TopI poisons like camptothecin trap TopI on DNA, leading to accumulated supercoiling and R-loops, which then cause lethal DNA breaks in rapidly dividing cells [92]. This is one mechanism by which TopI inhibitors kill cancer cells, especially those with defective R-loop repair. Targeting TCR in cancer is also of interest [93]. Tumors with proficient TCR can quickly repair transcription-blocking DNA lesions, which might reduce the efficacy of drugs like cisplatin or UV-mimetic therapies [94]. Therefore, inhibitors of TCR components could sensitize tumors to such treatments. While no specific CSB or CSA inhibitors are in clinical use, there is evidence that inhibiting transcription itself during therapy, for example, using CDK9 (cell cycle-dependent kinase 9) inhibitors to pause transcription globally can make DNA damage more toxic to cancer cells by preventing TCR, akin to a transcription-coupled sensitization strategy [95]. Another approach could be to target the proteolysis pathways [96] since TCR involves ubiquitination of RNAPII and CSB; modulating ubiquitin–proteasome activity might influence outcomes, though this would be quite non-specific.

### 6.3. Exploiting R-Loop Vulnerabilities with Synthetic Lethality

Certain cancers exhibit an inherent dependency on R-loop tolerance pathways due to underlying mutations [97]. For instance, as noted, cancers with spliceosome gene mutations common in myelodysplastic syndromes and some leukemias accumulate R-loops and activate the ATR checkpoint kinase to cope with the ensuing replication stress [82]. In these cells, ATR signaling is critical for survival. Indeed, preclinical studies have shown that spliceosome-mutant cancer cells are selectively sensitive to ATR inhibitors; the drugs cause excessive DNA damage and cell death in those cells, a phenotype that can be rescued by overexpression of RNase H1, proving R-loops were the cause of the vulnerability [82]. This suggests a synthetic lethal approach: target ATR or other genome stability pathways in cancers with high R-loop load to induce catastrophic failure of repair. ATR inhibitors are in trials for various tumors [98], and R-loop levels or spliceosomal mutations could be biomarkers to predict responder populations [99]. Similarly, cancers deficient in Fanconi anemia proteins or BRCA1 rely on alternate R-loop repair mechanisms [100]. Therefore, they might be particularly vulnerable to helicase inhibitors [45]. Paradoxically, inhibiting a helicase like DDX or SETX in such a cancer could create an unmanageable level of R-loops, specifically killing the cancer cells, while normal cells with intact pathways handle it [101]. Although developing small-molecule inhibitors of specific helicases is challenging due to large protein–nucleic acid interfaces [102], some efforts are underway to find compounds that interfere with R-loop resolvases [103]. The flip side is using R-loop stabilizing molecules to trap cancer cells: for example, bisquinoline compounds or G-quadruplex stabilizers might increase R-loops at gene promoters, hampering oncogene transcription [104]. Careful tuning would be needed to avoid harming normal cells, but the differential reliance on R-loop repair between cancer and normal cells could be exploited.

### 6.4. Therapeutic Targeting of TCR in Progeria and Neurodegeneration

Beyond rare syndromes, there is speculation that boosting TCR might ameliorate aspects of aging or neurodegeneration [105]. As we age, DNA damage accumulates in neurons and other non-dividing cells [106], potentially overwhelming TCR capacity. Pharmacological upregulation of TCR could be envisioned as small molecules that increase CSB expression or enhance TFIIH recruitment [107]. One interesting angle is the connection of TCR with the transcription of mitochondrial genes and general transcription stress in neurodegeneration [108]. While no such drug is available yet, some studies are exploring modulators of the NAD^+^–PARP1 axis since PARP1 can prevent transcription restart after damage [109]. Hence, PARP inhibitors might indirectly promote restart in some contexts. Also, anti-inflammatory approaches can be indirectly relevant: if R-loop accumulation triggers immune responses as in Aicardi–Goutières, where cGAS/STING may sense DNA:RNA hybrids, then anti-inflammatory drugs or nucleic acid metabolism enhancers might help [110].

## 7. Future Directions

The field is moving toward a more integrative understanding of transcription-associated genome stability. Future research will undoubtedly uncover new players that link TCR and R-loops, for example, how chromatin modifications influence the choice between resolving a stall via TCR versus letting an R-loop form. High-resolution live-cell imaging of transcription, repair, and R-loops simultaneously is an exciting frontier [111], which could visualize the cascade from RNAPII stalling to either repair or R-loop formation in real-time. Such studies will guide therapeutic timings. On the therapeutic side, we may see combination therapies that pair DNA-damaging agents with inhibitors of R-loop repair to selectively kill cancer cells [112] or combinatorial gene therapies that co-deliver a missing repair enzyme and an RNA processing factor to neurodegenerative patients to synergistically reduce damage [113]. Finally, the overlap of R-loops with other phenomena like trinucleotide repeat expansion disorders [114], viral infections which can produce R-loops or R-loop-like hybrids [115], and the role of R-loops in stem cell differentiation [116] are all active areas. Each could yield insights that translate to novel treatments, for instance, controlling R-loops to influence cell fate or to improve the efficacy of antiviral responses.

In conclusion, TCR and R-loops are two sides of the transcriptional maintenance coin: one repairs DNA to keep transcription moving, and the other is a transcription by-product that needs regulation. Both are vital for preventing transcription-associated genome instability. Ongoing research and technological advances are not only unraveling their complex interplay but also pointing towards ways to harness this knowledge to combat human disease. By targeting the mechanisms that cells normally use to safeguard transcription, we can envision therapies that either reinforce those safeguards when they fail, as in genetic diseases or disable them in a targeted way to attack pathological cells, as in cancer. In the coming years, significant progress will be made in translating the biology of TCR and R-loops into clinical interventions, making this an exciting frontier in molecular medicine.

## Figures and Tables

**Figure 1 ijms-26-03744-f001:**
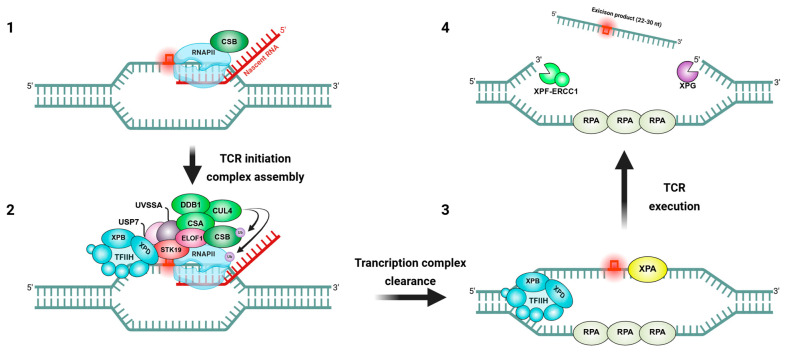
The Stepwise process of TCR. (**1**) Lesion recognition and RNAPII stalling: when bulky DNA damage is encountered on the transcribed strand, RNAPII stalls at the site. CSB, an ATP-dependent DNA translocase, is rapidly recruited and initiates the TCR response. (**2**) CSA is recruited and, with DDB1 and CUL4, forms an E3 ubiquitin ligase complex that ubiquitylates CSB (and possibly RNAPII), marking the stalled transcription complex for processing. UVSSA is also recruited to stabilize CSB and support repair factor assembly, while additional factors such as ELOF1 and STK19 promote TFIIH recruitment and enhance TCR efficiency. (**3**) DNA unwinding and damage verification: TFIIH, containing XPB and XPD helicases, unwinds the DNA around the lesion. XPA verifies the damage, and RPA binds single-stranded DNA to maintain stability. (**4**) Excision of the damaged DNA segment: the endonucleases XPG (3′ incision) and XPF–ERCC1 (5′ incision) remove a 22–30 nucleotide segment that carries the DNA lesion, generating a single-stranded gap. The figure was created with BioRender.com, accessed on 15 March 2025.

**Figure 2 ijms-26-03744-f002:**
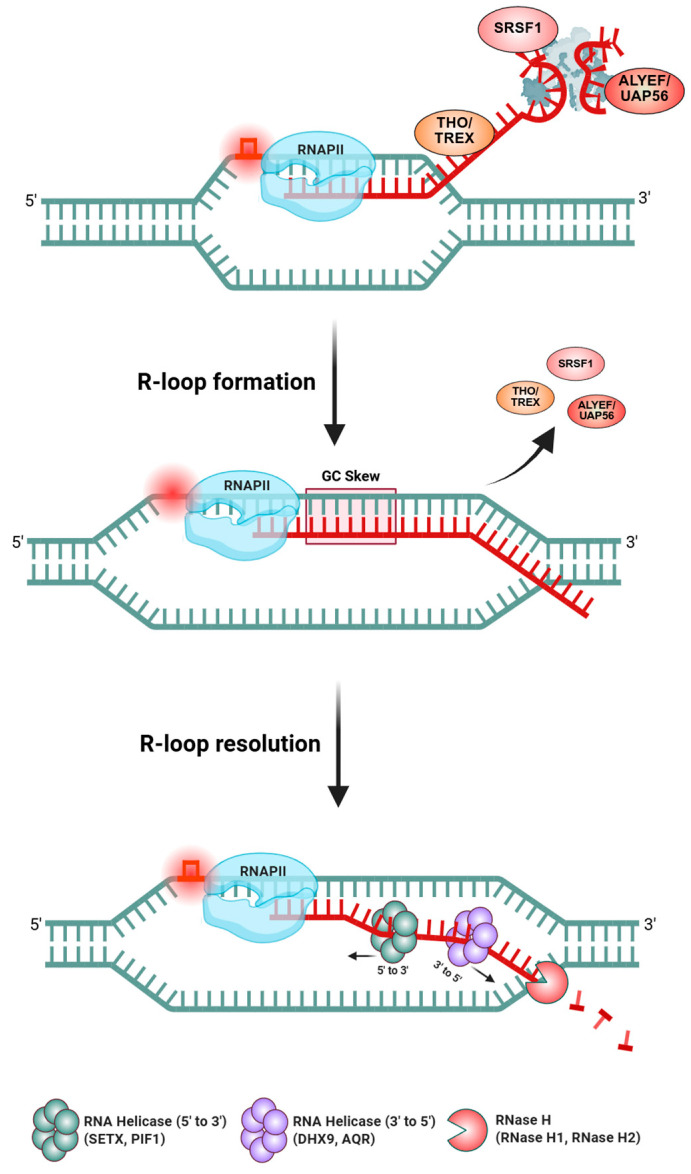
R-loop formation and resolution during transcription. Under normal conditions, nascent RNA is bound and protected by THO/TREX, SRSF1, and ALYREF/UAP56, preventing it from re-annealing to the DNA behind RNAPII. However, factors such as transcriptional pausing, torsional stress, or GC-skewed DNA sequences can drive the RNA to invade the DNA duplex and form an R-loop. Once formed, R-loops can be resolved by RNA helicases such as SETX and PIF1 (5′→3′ direction) or DHX9 and AQR (3′→5′ direction), depending on the polarity of the RNA strand. RNase H1 and H2 further degrade the RNA component of RNA:DNA hybrids to ensure complete resolution and maintain genome stability. The figure was created with BioRender.com, accessed on 15 March 2025.

**Figure 3 ijms-26-03744-f003:**
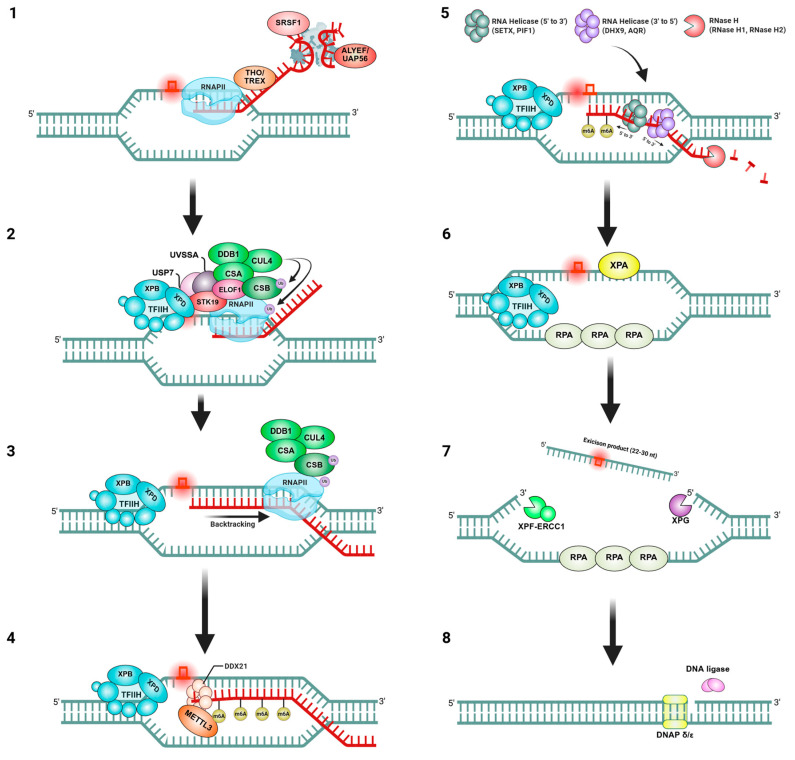
Bidirectional interplay between TCR and R-loops This schematic illustrates TCR, a specialized DNA repair pathway that resolves transcription-blocking lesions encountered by RNAPII, while also highlighting how R-loops are regulated during this process. (**1**) Transcription and R-loop prevention: during normal transcription, nascent RNA is bound by THO/TREX, SRSF1, and ALYREF/UAP56, preventing its rehybridization with the DNA template and thereby suppressing R-loop formation behind elongating RNAPII. (**2**) Lesion recognition and TCR initiation: upon stalling of RNAPII at a DNA lesion, CSB and CSA (with DDB1–CUL4) are recruited to the site, along with UVSSA, USP7, STK19, and ELOF1. These factors initiate the TCR process. (**3**) Backtracking and DNA exposure: RNAPII is backtracked under the guidance of CSB/CSA, exposing the DNA damage for processing. TFIIH is also positioned at the lesion site in preparation for unwinding. (**4**) m6A modification by DDX21–METTL3 axis: the helicase DDX21 is recruited to the lesion-proximal R-loop, where its helicase activity facilitates access of METTL3 to nascent RNA. This allows N6-methyladenosine (m6A) modification near the damage, which may modulate RNA stability and influence R-loop resolution. (**5**) R-loop resolution: if R-loops persist, RNA helicases (SETX, PIF1 in the 5′→3′ direction; DHX9, AQR in the 3′→5′ direction) unwind RNA:DNA hybrids. RNase H1/H2 cleave the RNA strand within the hybrid to fully resolve the structure and prevent genome instability. (**6**) Damage verification and unwinding: TFIIH (XPB, XPD) unwinds the DNA duplex at the damage site, while XPA confirms the presence of a lesion. RPA stabilizes the single-stranded DNA region that is generated. (**7**) Dual incision and gap formation: endonucleases XPF–ERCC1 (3′ side) and XPG (5′ side) excise the damaged oligonucleotide (22–30 nt), generating a single-stranded DNA gap. (**8**) Repair synthesis and completion: DNA polymerase δ/ε fills the gap, and DNA ligase seals the nick to complete the repair. Transcription can then resume on the repaired DNA template. The figure was created with BioRender.com, accessed on 15 March 2025.

**Figure 4 ijms-26-03744-f004:**
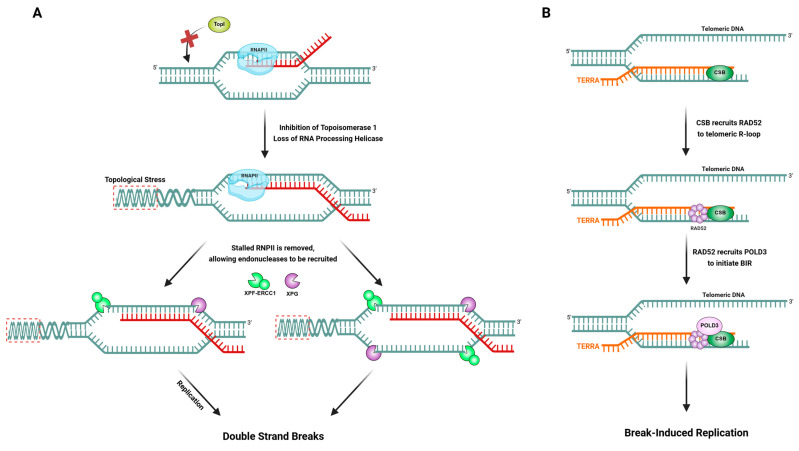
Distinct fates of TCR–R-loop interplay. (**A**) When TopI activity is inhibited, or helicases like SETX and AQR are lost, R-loops accumulate behind elongating RNAPII and can stall transcription. These transcriptional blocks may be mistaken as DNA lesions, prompting the recruitment of CSB and its associated endonucleases, XPF–ERCC1 and XPG. The stalled RNAPII is removed either through eviction or degradation allowing XPF–ERCC1 and XPG to be recruited. Unlike classical NER, which typically induces single-strand DNA gaps, R-loop processing by these enzymes may involve dual incisions on both the transcribed and non-transcribed strands. This could directly result in DSBs or, alternatively, produce nicks or gaps that disrupt replication fork stability during the S phase. (**B**) Conversely, at telomeres where DNA damage occurs, R-loops formed by TERRA play a beneficial role. These structures recruit the TCR factor CSB, which facilitates the localization of RAD52 to the R-loop. RAD52 subsequently recruits POLD3, initiating BIR, a recombination-based repair mechanism essential for telomere maintenance. In this context, R-loops act not as a source of instability but as functional intermediates that guide repair factors toward genome-preserving outcomes. The figure was created with BioRender.com, accessed on 15 March 2025.

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
