# Peer review of "Transcription-Coupled Repair and R-Loop Crosstalk in Genome Stability"

_ijms, 2025, doi:10.3390/ijms26083744_

Round 1
Reviewer 1 Report
Comments and Suggestions for Authors
The review by Jeon and Kang addresses the topic of complex relationship between TCR and R-loop and their collective impact on genome stability. It is a very interesting subject that should resonate with researchers in many different fields. It highlights the importance of homeostasis of R-loop formation and the often difficult strategic decision a cell must make to balance transcription progression and DNA repair efficiency in preserving genome integrity. Overall, the review is an engaging read and provides some important insights. My main comments are that some descriptions of the R-loop topology are incorrect and that the model figures should be improved.
Notation of DNA relative to the R-loop: In Figure 1 panel 1, the authors depict an R-loop with the top DNA strand being the template strand, the RNA Pol II traveling from the right to the left, and a CSB binding to the left side of Pol II. In the text (line 72), the authors stated “ CSB…binds upstream of RNAPI”. This is confusing because CSB is depicted downstream of the Pol II machinery with respect to transcription direction. Based on Ramadhin et al. (reference 36), CSB indeed binds upstream of Pol II. In the diagram here, CSB should be placed on the right side of Pol II. Please see Figure 1A of reference 19 (Xu et al. 2018) for the correct depiction of “upstream” and “downstream” regions relative to Pol II transcription and modify all figures and text accordingly. In Figure 3, the entire TCR complex should be upstream of Pol II and TFIIH being downstream, based on the model in Ramadhin et al.
Directionality of RNA helicases: In Figure 2, RNA helicases (SETX, DHX9, PIF1) and RNase H all translocate 5’ to 3’ on the RNA. At least the helicases have different polarity, with SETX and PIF1 being 5’ to 3’ and DHX9 being 3’ to 5’. It would be important to delineate these enzymes based on this attribute.
Missing key elements in TCR-R-loop interplay: In Figure 4A, authors talked about how R-loops can become DSB due to incisions by XPF-ERCC1 and XPG. It is important to show where the incisions occur to generate a DSB. In Figure 4B, please show the position of telomere relative to the R-loop and where BIR takes place.
Suggested citation: When discussing R-loop accumulation in diseases, particularly in the FXS, it is important to cite Chakraborty A et al, 2020 where authors showed that FXS cells suffer from R-loop-induced DSBs genome-wide, not simply the CGG repeat locus in FMR1.
Author Response
Notation of DNA relative to the R-loop: In Figure 1 panel 1, the authors depict an R-loop with the top DNA strand being the template strand, the RNA Pol II traveling from the right to the left, and a CSB binding to the left side of Pol II. In the text (line 72), the authors stated “ CSB…binds upstream of RNAPI”. This is confusing because CSB is depicted downstream of the Pol II machinery with respect to transcription direction. Based on Ramadhin et al. (reference 36), CSB indeed binds upstream of Pol II. In the diagram here, CSB should be placed on the right side of Pol II. Please see Figure 1A of reference 19 (Xu et al. 2018) for the correct depiction of “upstream” and “downstream” regions relative to Pol II transcription and modify all figures and text accordingly. In Figure 3, the entire TCR complex should be upstream of Pol II and TFIIH being downstream, based on the model in Ramadhin et al.
Reply:
We sincerely thank the reviewer for the valuable feedback regarding the spatial orientation of the transcription complex and associated factors. In accordance with the models proposed by Ramadhin et al. and Xu et al. (2018), we have revised Figure 1 (panel 1) so that CSB is now positioned upstream of RNAPII, correctly reflecting the direction of transcription from left to right. Additionally, Figure 3 has been modified to illustrate the TCR complex assembly upstream of RNAPII, while TFIIH is now depicted downstream, consistent with the established model of transcription-coupled repair. These corrections clarify the topological relationship between RNAPII, CSB, and other TCR factors, and resolve the previously noted inconsistency between the figure and the main text.
Directionality of RNA helicases: In Figure 2, RNA helicases (SETX, DHX9, PIF1) and RNase H all translocate 5’ to 3’ on the RNA. At least the helicases have different polarity, with SETX and PIF1 being 5’ to 3’ and DHX9 being 3’ to 5’. It would be important to delineate these enzymes based on this attribute.
Reply:
We have carefully revised Figure 2 to accurately represent the translocation polarity of each RNA helicase. Specifically, SETX and PIF1 are now indicated to translocate in the 5′ to 3′ direction, whereas DHX9 and additionally AQR are shown to move in the 3′ to 5′ direction along the RNA strand. Directional arrows have been added to each enzyme to facilitate visual differentiation. We have also updated Figure 3 to reflect these corrections consistently across the manuscript. These revisions enhance the accuracy and clarity of our model depicting R-loop resolution.
Missing key elements in TCR-R-loop interplay: In Figure 4A, authors talked about how R-loops can become DSB due to incisions by XPF-ERCC1 and XPG. It is important to show where the incisions occur to generate a DSB. In Figure 4B, please show the position of telomere relative to the R-loop and where BIR takes place.
Reply:
To enhance the visual clarity of the R-loop processing mechanism and its pathological consequences, we have revised Figure 4A to explicitly mark the incision sites mediated by XPF–ERCC1 and XPG on the DNA. These modifications help illustrate how bidirectional cleavage can lead to double-strand breaks (DSBs), particularly under conditions of Topoisomerase I inhibition or loss of RNA processing helicases, which promote R-loop accumulation.
In addition, Figure 4B has been updated to clearly indicate the telomeric location where TERRA hybridizes and to more explicitly illustrate the Break-Induced Replication (BIR) process. The stepwise recruitment of CSB, RAD52, and POLD3 is depicted to emphasize the strand-specific role of TERRA and the associated repair machinery. These revisions are expected to improve readers’ understanding of the physiological and pathological outcomes of TCR–R-loop interplay.
Suggested citation: When discussing R-loop accumulation in diseases, particularly in the FXS, it is important to cite Chakraborty A et al, 2020 where authors showed that FXS cells suffer from R-loop-induced DSBs genome-wide, not simply the CGG repeat locus in FMR1.
Reply:
As suggested by the reviewer, we have cited Chakraborty A et al., 2020 in Section 5.2 to acknowledge the genome-wide nature of R-loop–induced DNA double-strand breaks in Fragile X Syndrome (FXS). The following sentence has been added to the paragraph discussing FXS:
“Importantly, recent evidence suggests that R-loop–induced DNA double-strand breaks (DSBs) in FXS are not confined to the CGG repeat locus but can occur across the genome [71].”
This addition highlights the broader impact of R-loop dysregulation in FXS pathophysiology, beyond locus-specific instability.
Reviewer 2 Report
Comments and Suggestions for Authors
In the manuscript, the authors introduced the transcription-coupled repair and R-loop, mainly including their detail mechanism, the associated proteins (including their proteins), and most important the connection between transcription-couples repair and R-loop. And finally, the authors introduced some treatments associated with transcription-coupled repair and R-loop (even though present clinical treatments are rarely present as far as I am concerned). And finally, the charming is that the authors provide reasonable perspective (new player et al.). This is one of the reviews which I have read clearly introduced transcription-coupled repair and R-loop. And the work helps readers improve the understanding the DNA repair and DNA metabolism. What is more, the structure and the writing of the manuscript are both good. Therefore, the manuscript is recommended to be accepted by the International Journal of Molecular Sciences. Here are some minor concerns for the authors:
1) For the title, as the authors used one connector, the titled should be revised as “Transcription-coupled Repair and R-Loop Crosstalk in Genome Stability”.
2) In the Abstract section, the authors are suggested to use the sentence format “not only…but also…” in the 3rd
3) In the Keywords section, I think “Synthetic lethality” and “Helicases” should be deleted as they are not the topics of the manuscript. What is more, “Cancer treatment” and/or “Therapeutic methods” should be added to this section.
4) When introducing others’ work, the authors tend to use “Author et al”. Such typing is recommended to revise as “Author et al.” (add one period).
5) In section 3, when introducing the formation of R-loop, as recently CRISPR-Cas-associated gene editing also mentioned the formation of R-loop (10.3390/molecules30040947). Therefore, I think such R-loop formation should be mentioned and added in line 155 to line 158 of the manuscript.
6) In section 4, as the authors mentioned the interplay between TCR and R-loop including the interaction of TCR to R-loop and the interaction of R-loop to TCR. Therefore, the authors are encouraged to add two meaningful subtitles before these two kinds of summarize.
7) In section 5, the authors are also encouraged to add subtitled.
Author Response
1) For the title, as the authors used one connector, the titled should be revised as “Transcription-coupled Repair and R-Loop Crosstalk in Genome Stability”.
Reply:
We sincerely thank the reviewer for the constructive suggestion regarding the manuscript title. In response, we have revised the title to:
“Transcription-coupled Repair and R-loop Crosstalk in Genome Stability” to ensure clarity and consistency in conjunction usage.
2) In the Abstract section, the authors are suggested to use the sentence format “not only…but also…” in the 3rd
Reply:
We have revised the third sentence in the Abstract to incorporate the suggested structure:
“In contrast, R-loops—RNA–DNA hybrid structures formed co-transcriptionally—play not only regulatory roles in gene expression and replication but can also contribute to genome instability when persistently accumulated.”
3) In the Keywords section, I think “Synthetic lethality” and “Helicases” should be deleted as they are not the topics of the manuscript. What is more, “Cancer treatment” and/or “Therapeutic methods” should be added to this section.
Reply:
We have removed “Synthetic lethality” and “Helicases” from the Keywords section and added “Cancer treatment” to better reflect the manuscript’s content.
4) When introducing others’ work, the authors tend to use “Author et al”. Such typing is recommended to revise as “Author et al.” (add one period).
Reply:
We appreciate the reviewer’s attention to detail. As recommended, we have carefully revised all in-text citations to ensure consistent use of “et al.” with a period.
5) In section 3, when introducing the formation of R-loop, as recently CRISPR-Cas-associated gene editing also mentioned the formation of R-loop (10.3390/molecules30040947). Therefore, I think such R-loop formation should be mentioned and added in line 155 to line 158 of the manuscript.
Reply:
We have added the following sentence to Section 3 to acknowledge the recent findings regarding R-loop formation in CRISPR/Cas gene editing:
“Notably, recent studies have also reported that the CRISPR/Cas gene editing system can induce R-loop formation during target DNA recognition and cleavage, as part of its mechanism for creating site-specific double-strand breaks [43].”
This addition highlights the broader biological relevance of R-loop structures beyond endogenous transcription, extending to genome editing technologies.
6) In section 4, as the authors mentioned the interplay between TCR and R-loop including the interaction of TCR to R-loop and the interaction of R-loop to TCR. Therefore, the authors are encouraged to add two meaningful subtitles before these two kinds of summarize.
Reply:
We have divided Section 4 into clearly delineated subsections, each with a descriptive subtitle. These subtitles distinguish the directionality of the interplay—how TCR affects R-loops, and how R-loops modulate TCR—to help guide the reader through the bidirectional relationship and enhance the logical structure of the section.
7) In section 5, the authors are also encouraged to add subtitled.
Reply:
In line with the reviewer’s recommendation, we have revised Section 5 by adding appropriate subtitles to each subsection. The new headings reflect the distinct pathological outcomes associated with TCR or R-loop dysregulation, including Cockayne Syndrome, neurodegeneration, and cancer, thereby improving the clarity and navigability of the section.